# A Suggestion of Converting Protein Intrinsic Disorder to Structural Entropy Using Shannon’s Information Theory

**DOI:** 10.3390/e21060591

**Published:** 2019-06-14

**Authors:** Hao-Bo Guo, Yue Ma, Gerald A. Tuskan, Hong Qin, Xiaohan Yang, Hong Guo

**Affiliations:** 1Department of Computer Science and Engineering, SimCenter, University of Tennessee, Chattanooga, TN 37403, USA; 2Department of Biochemistry and Cellular and Molecular Biology, University of Tennessee, Knoxville, TN 37996, USA; 3Biosciences Division, Oak Ridge National Laboratory, Oak Ridge, TN 37831, USA; 4Department of Biology, Geology, and Environmental Science, University of Tennessee, Chattanooga, TN 37403, USA

**Keywords:** intrinsically disordered proteins, entropy, information

## Abstract

We propose a framework to convert the protein intrinsic disorder content to structural entropy (*H*) using Shannon’s information theory (IT). The structural capacity (*C*), which is the sum of *H* and structural information (*I*), is equal to the amino acid sequence length of the protein. The structural entropy of the residues expands a continuous spectrum, ranging from 0 (fully ordered) to 1 (fully disordered), consistent with Shannon’s IT, which scores the fully-determined state 0 and the fully-uncertain state 1. The intrinsically disordered proteins (IDPs) in a living cell may participate in maintaining the high-energy-low-entropy state. In addition, under this framework, the biological functions performed by proteins and associated with the order or disorder of their 3D structures could be explained in terms of information-gains or entropy-losses, or the reverse processes.

## 1. Introduction

Protein structures have played a central role in molecular biology ever since the “lock-and-key” model for enzymatic catalysis [1], along with the sequence-structure-function dogma [2] and the protein folding landscape theory [3]. However, many proteins from different organisms have been found to lack stable, well-folded tertiary structures in their native states. These proteins are termed the intrinsically disordered proteins (IDPs) or intrinsically disordered protein regions (IDPRs) [4]. The term “intrinsic” here means that the quantity is encoded in the protein primary amino acid sequence, on the same ground as the IDP denominator [5]. The abundance of IDPs and IDPRs, and the vital roles they play in cellular organisms and viruses, has been appreciated in a plethora of publications during the past two decades [6,7]. Despite the extensive discussions of IDPs in literature, “paradoxes” seem to present in IDP structures and functions, including reduced structural content versus enhanced structural heterogeneity, simplified folding landscapes versus complexified functionalities, etc. [8] 

The intrinsic disorder content of each residue in a protein can be predicted from its primary amino acid sequence. It is now well understood that the IDPs lack the hydrophobic cores necessary for successful folding [9] and that the charged residues (R, K, H, E, and D) are disorder-promoting [10]. An early study showed the association between the ratios of the numbers of charged over hydrophobic residues and the disorder content for a relatively small set of proteins [11]; however, this oversimplified approach was not predictive for larger protein sets. As representative present algorithms, the Predictor Of Natural Disordered Regions (PONDR) family predictors use artificial neural networks to integrate attributors, including the amino acid composition, sequence complexity, hydropathy, and net charge—and they yield reasonably accurate intrinsic disorder predictions [12]. 

Developing accurate intrinsic disorder predictors is an endless path [13], considering the astronomical size of the protein world—e.g., 20^100^ different proteins of 100 amino acids—and the crowded and dynamic cellular environments surrounding the proteins. Quality assessment has been performed among different protein disorder predictors [14,15], which adopt the same standard for the intrinsic disorder score, representing the probability of being disordered—i.e., a fully ordered residue scores 0 (0%) and a fully disordered residue scores 1 (100%), respectively. The need for a unified guideline to report a protein intrinsic disorder has also been demanded [16]. In intrinsic disorder interpretations, ambiguities exist for both amino acids and proteins. For example, if an amino acid has an ID score of 0.5, is it an ordered or disordered residue? Similarly, for a protein with 50% disordered residues, is it an ordered or disordered protein? 

In the present work, we suggest a conversion to the intrinsic disorder contents of the residues in the proteins into entropy contents, similar to the process developed by Shannon over 70 years ago [17]. In information theory (IT), the Shannon entropy uses a scoring system very similar to the ID predictors: A fully determined state scores 0 and a fully uncertain state scores 1, respectively. It is clear that the intrinsic disorder contents consist of the information that the protein carries. However, a closer examination must be taken to determine if the intrinsic disorder content could be treated as the disorder entropy, or structural entropy. Below we will show that the protein intrinsic disorder (*D*) cannot be regarded as the structural entropy (*H*) directly. In addition, a means for converting *D* to *H* was proposed that provides a conceptual framework that quantitatively defines the structural information carried by a protein in vitro.

Hereafter, we will use structural entropy—or simply entropy—in accordance with Shannon’s insight that the information and entropy serve as measures of each other: Information is negative entropy, or negentropy (Equation (2) and see below). Note that the term “structural” defines a *meaning* for the information; however, the semantic aspect of information was originally discarded by Shannon [17] and later by some molecular biologists [18]. Still, other biologists argued that meanings or purposes are crucial to biological information [19,20]. Discussions on syntactics versus semantics in IT can be found in other publications [21], but are beyond the scope of this paper.

A fundamental difference between *D* and *H* is that the structural entropy, *H*, is an additive quantity, whereas the ID score, *D*, is not (see the Appendix A). That is, adding up structural entropies of all residues leads to the structural entropy of the entire protein, whereas we cannot infer the ID score of a protein by adding up the ID scores of all its residues. We suggest that the (intrinsic) structural entropy either of a single residue in the protein or of the entire protein can be directly calculated from a disorder predictor. Our suggestion indicates that it is feasible to quantitatively compare structural entropies of two proteins, regardless of their respective disorder percentages, which are rough and qualitative measures of the protein disorder.

Our suggestion may also help to understand protein structures and functions in the cells quantitatively and provide an easy answer to the paradoxes mentioned above: Reduction in information is equivalent to an increase in entropy or complexity. Under this framework, the processes in a living cell might be quantitatively viewed as the procedures for maintaining the low entropy levels.

## 2. Shannon’s Equation: Structural Entropy (*H*), Information (*I*), and Capacity (*C*)

Shannon’s equation [17] implies:(1)$$H\left(X\right)=-K\sum_{i=0}^{n}p_{i}\log p_{i},\;C\left(X\right)=H_{max}\left(X\right)=H\left(X\right)+I\left(X\right)$$
where *H*(*X*) is defined as the entropy—or uncertainty—of *X*, which is any source of information with the number of states *n* and *p_i_* is the probability of the *i-*th state of *X*. *K* is an arbitrary scaling factor, which means that the base of the logarithm function could vary depending on the unit of the entropy. For example (set *K* = 1), a base of two corresponds to the unit of the binary digit (bit), and the natural base (ln) leads to the unit of the natural unit (nat). The maximum value of *H*(*X*) *H*_max_(*X*), determines the upper limit or the capacity of *X*, *C*(*X*), which Shannon considered the channel capacity for communications [17]. The second half of Equation (1), though not explicitly stated by Shannon, indicates that, for a given capacity *C*(*X*), the entropy *H*(*X*) could serve as a measure of the information *I*(*X*) that one has received—that is,
(2)ΔI(X)=−ΔH(X).

As a consequence, the information change Δ*I* was termed negentropy (negative entropy) [22], which is what Maxwell’s demon receives to generate order out of disorder [23].

To connect the intrinsic disorder contents obtained by regular disorder predictors, the structural entropy of a protein *X* with a length of *L* is defined as:(3)$$H\left(X\right)=\overset{L}{\underset{i=1}{\sum }}-x_{i}\log_{2}x_{i}-\left(1-x_{i}\right)\log_{2}\left(1-x_{i}\right),\;\;\;x_{i}=d_{i}/2\;$$
where *d_i_* is the intrinsic disorder content of the *i-*th residue. 

In the fully disordered state (*d_i_* = 1 for all residues), the protein exhibits the maximal entropy, expressed as the structural capacity of:(4)$$C\left(X\right)=H_{max}\left(X\right)=\log_{2}2^{L}=L$$
which carries zero structural information *I*(*X*). When all residues are fully ordered (*d_i_* = 0 for all residues), *H*(*X*) = 0 since $\underset{x\to 0}{\lim}x\log x=0$, and the *I*(*X*) reaches the maximum of *L*. Normally, 0 ≤ *d_i_* ≤ 1 and 0 ≤ *H*(*X*) [or *I*(*X*)] ≤ *L*. The criteria for using the logarithmic measure suggested by Shannon [17] then holds. In Equation (3), each residue is in a two-state system, with the probabilities *x_i_* in one state and (1 – *x_i_*) in the other state, and the total number of states is *n* = 2*^L^*. More details and potential alternatives to this approach have been summarized in the Appendix A.

Equation (4) indicates that the structural capacity (*C*) of a protein is its amino acid (AA) sequence length (L). In addition, it had been reported that *C* (protein length *L* was used in the original report, but since it is equivalent to *C*, only *C* will be used hereafter to prevent unexpected confusion) has an exponential distribution or a gamma distribution [24]. We used three models, the exponential, gamma, and power law distributions, to fit the structural capacities arranged in hierarchical ranks from the smallest to the largest (Appendix A). The exponential model provides the best overall fit to the structural capacities in all proteomes studied in the present work. The gamma model, in some cases, produces a better fit than the exponential model, but it fails to recapture the small-*C* regions for the two animal proteomes (human and fruit fly). The power law model fits the small-*C* but fails at the large-*C* regions. Therefore, the exponential model is the best choice here to describe the protein structural capacity distributions, exemplified in Figure 1A,D. Fittings of all proteomes studied here are summarized in Appendix A.

An exponential fit of *C* implies a linear fit of log*C* (with *C*s ranked hierarchically), which may therefore be a better measure of the protein-capacity distributions in proteomes [24]. Interestingly, with a bin size of Δlog_2_*C* = 1, Gaussian distributions can be observed in log*C* of all proteomes exemplified in Figure 1B,D. The underlying mechanisms of this Gaussian distribution have not been discussed in the literature; however, the distribution density can also be understood as the potential of mean force, or free energy, using the relationship:(5)$$\Delta G=-RT\ln\rho_{i}$$
where *R* is the gas constant, *T* is the temperature, and *ρ_i_* is the density of proteins at the *i-*th histogram of protein capacities. A preferential free-energy well (insets in Figure 1B,D) may present in the proteome, confining the protein structural capacity distributions. The locations and depths of this well vary among organisms: For the human proteome, the well depth is 3.16 kcal/mol centered at *C =* 592, whereas for *JCVI-Syn3.0* the well depth is 1.70 kcal/mol centered at *C* = 401. 

## 3. Entropy-to-Information Ratio, *R*

The extremely dynamic environment in the cell renders difficulties in detection or prediction of the changes of the structural entropies and structural information. However, the sum of both quantities equals the structural capacity that is the same as the AA sequence length, and the information gain always equals the entropy loss and vice versa. Hence, it is useful to know the relationships between the intrinsic structural entropies and the information that can be derived (Equation (3)) from the residual disorder contents, which, in turn, can be predicted based on the protein amino acid sequences. 

Here, we define the entropy-to-information ratio, *R*, as *R* = *H*:*I*. *R* ranges from 0 (for a fully ordered protein with *H* = 0 and *I* = *C*) to ∞ (for a fully disordered protein with *H* = *C* and *I* = *0*). When *R* > 1, the protein is entropy rich (*H* > *I*), and when *R* < 1, the protein is information rich (*H* < *I*). 

We noticed no apparent correlation between *C* and *R* for all proteomes—e.g., the Pearson correlation coefficients between *C* and *R* is –0.03 for both *H. sapiens* and *Arabidopsis thaliana*, and a similar trend can be found in other proteomes studied in the present work. This lack of correlation permits the construction of a protein space with two attributes *C* and *R*: The former represents the sum and the latter represents the ratio of the structural entropy (*H*) and information (*I*), respectively. This space is termed the protein *CR*-space hereafter. In the *CR-*space, *C_i_* sets the upper limit and *R_i_* gives the intrinsic ratio between the two quantities of the total *H_i_* and *I_i_* of the *i-*th protein *P_i_*. Based on the Gaussian distribution in both log_2_*C* and log_2_*R*, the protein distributions in the *CR-*space are represented in Figure 2, for proteins expressed in representative prokaryotes (bacteria and archaea, Figure 2A), eukaryotes (Figure 2B), viruses (Figure 2C), and datasets collected from the DisProt database [25] and protein data bank (PDB) [26] (Figure 2D). 

Table 1 summarized the protein distributions in the *CR-*space of all proteomes and datasets, and more discussions will be given as follows.

### 3.1. Prokaryotes vs. Eukaryotes

Figure 2A,B show the protein distributions of prokaryotic and eukaryotic proteomes, respectively. The total number of proteins vary from the smallest synthesized bacterium *JCVI-Syn3.0* that has only 438 proteins [27] to the monocot plant *Oryza*
*sativa* (rice) that has 48,782 proteins [28,29]. A significant difference between prokaryotes and eukaryotes is the overall *R* (*ΣH:**ΣI*) of the proteomes: Values for prokaryotes range from 1.29 (*Nanoarchaeum*
*equitans*) to 1.57 (*Ignicoccus*
*hospitalis*) and values for eukaryotes range from 2.10 (*Amborella*
*trichopoda*) to 2.53 (*H. sapiens*). 

Table 1 also shows the extreme *C* and *R* of all proteomes. The most distinct difference between prokaryotes and eukaryotes is the number of totally disordered proteins that have *R_max_* = ∞, which is induced by *I* = 0 (*H* = *C*) of these proteins. In the prokaryotes, only one such protein can be found in *Lokiarchaeum* [30], which exhibits strong potential to represent the ancestral host archaeon that accidentally swallowed the bacterium (*Rickettsiale* being a potential relative [31]), which fortunately turned into a symbiont and eventually evolved to the mitochondria of eukaryotic cells. All eukaryotic proteomes, however, contain fully disordered proteins, from two in *Saccharomyces cerevisiae* to 37 in *H. sapiens*.

It should be mentioned that there are 25 seleno-proteins in the human proteome, which has an unusual type of residue of selenocysteine (one letter code U, or three letter code Sec), though the disorder contents of these proteins cannot be predicted by the current version of PONDR predictors (see the SM for a list of the human seleno-proteins). These proteins were not included in the final analysis of the current study.

### 3.2. Viruses

Figure 2C shows the protein distributions of two giant DNA viruses (giruses) and two RNA viruses. The *R*’s of viruses vary significantly from 0.96 (*human coronavirus*) to 2.09 (*Pandoravirus*). 

Although the RNA viruses generally contain less than 10 protein-encoding genes, the protein intrinsic disorder—i.e., structural entropy—may be reflective of immune evasion and virus-host interactions, especially for the *Ebola* virus, in which all proteins are entropy-rich and have *R* > 1 [32]. 

On the other hand, the giruses have proteomes similar to those of some cellular organisms in both total protein numbers and their distributions in the *CR*-space. For example, *ΣH:**ΣI* of *Mimivirus* is 1.56, close to that of the prokaryotes, and *ΣH:**ΣI* of *Pandoravirus* is 2.09, close to that of some of the eukaryotes, such as the basal angiosperm *A. trichopoda*. Moreover, *Pandoravirus* has a fully disordered protein with a capacity of 29.

### 3.3. Datasets

The present work focuses on the intrinsic disorder and omits the environmental cues, which will certainly have great impact on determinations of the structural information and entropies. In contrast, the proteins collected in the datasets (DisProt [25] and PDB [26]) have been validated (or somewhat biased) experimentally. The protein distributions of the selected datasets are shown in Figure 2D.

Not surprisingly, the proteins from the DisProt database exhibit the highest *ΣH:**ΣI* of 2.99. However, several proteins (8 out of 803, or 1%) have *R* < 1. This difference may be caused either by the protein disorder prediction methods used here or by inherent limits of the validation protocol of the database. The eight information-rich proteins from the DisProt database, as well as their sequences, have been listed in the SM. Apparently, the human Titin [33] protein (*C =* 34,350, *R* = 2.52) is collected in DisProt. However, a closer look finds that that protein in DisProt points to the nuclear magnetic resonance (NMR) structure (PDB entry 1BPV) of the type-1 module of the Titin protein (*C* = 112, *R* = 2.30). Nevertheless, the Titin protein does have an extremely long IDPR based on our calculation (Appendix A). 

We further analyzed the protein sequence sets from the PDB. After removing redundant entries, each sequence in the following datasets represents a unique sequence: NMR set for sequences of structures collected by solution NMR (8,476 entries), X-ray 1.5 set for sequences from X-ray crystallographic structures with resolution higher than 1.5 Å (5,842 entries), and X-ray 3.0 set for sequences from X-ray structures with resolution lower than 3.0 Å (10,636 entries). The NMR method largely limited the size of proteins with the majority of *C*s smaller than 200. However, the structures detected by NMR generally have higher flexibility, leading to larger *R’*s (*ΣH:**ΣI =* 2.65). The X-ray 1.5 and X-ray 3.0 sets have *ΣH:**ΣI* of 1.78 and 1.73, respectively, but no significant difference was observed between these two datasets. It seems that the structural entropies of protein structures deposited in the PDB are method sensitive but are not determined by the resolution of the crystallography. 

It is worth noting that many structures deposited in the PDB (especially those solved by X-ray crystallography) have missing residues. However, here we are interested in their sequences that contain the missing residues. We also noticed that there are fully disordered (i.e., *R* = ∞) proteins in all datasets from PDB (Table 1). It may not be surprising to solve the highly disordered proteins using NMR, especially for relatively small proteins. However, even the X-ray 1.5 set contains fully disordered proteins. We listed the fully disordered proteins with *C* > 20 in both X-ray 1.5 and X-ray 3.0 sets in the SM (Appendix A). First, these proteins are relatively short with the longest one (1JCD, X-ray 1.5 set) having 52 residues. It may be possible that the PONDR predictor did not perform well for these sequences, particularly the collagen sequences (Appendix A). On the other hand, all listed proteins either are subunits of big protein complexes or form oligomers (mostly homo-trimers), suggesting coupled folding and binding may help these IDPs attain the folded structures [34]. The mutual folding induced by the binding (MFIB) database has collected 205 IDPs from PDB (both X-ray and NMR structures), with their folding induced by binding [35]. None of the proteins (Appendix A) have been collected in the MFIB database, but several entries listed in Appendix A have also been collected in the intrinsically disordered proteins with extensive annotations and literature (IDEAL) [36] database. Our findings may help expand the collection of MFIB, IDEAL, and similar databases.

### 3.4. Random Sequences

We randomly generated 500 proteins with random capacities in the range [50,800]. Interestingly, this random set has a *ΣH:**ΣI* ratio of 1.020 (Appendix A), indicating that the total entropy and information are roughly 1:1. This ratio is smaller than those from cellular proteomes shown in Figure 2 and is close to the viral proteome of human coronavirus (Figure 2C), suggesting that when total structural entropy is found to be relatively larger than total structural information in cellular organisms, that relationship is not random.

## 4. Evolution of Protein Structural Entropies and Information

Based on the analysis of cellular organisms and viruses, it seems that the *ΣH:**ΣI* ratio may set a boundary at approximately 1.6 to 2.0 for separating prokaryotes and eukaryotes. In addition, if there were such a boundary, the two giant DNA viruses (giruses) studied here would be located at different branches, with *Mimivirus* close to or within the prokaryotic branch and *Pandoravirus* close to or within the eukaryotic branch, respectively.

Previously, by using the protein length *L* (structural capacity *C*) and the disorder content *D*, distributions of proteins in the space with attributes *L* and *D* have been used to reconstruct a phylogenetic tree that indicates intriguing evolutionary dynamics associated with the variations of *L* and *D* in the proteomes [37]. A similar approach is used here to study protein distributions in the *HI-*space. 

The *CR*-space is split into 5 × 4 blocks (Figure 3A and Table 2). The gene densities [38] at different blocks are then calculated, and the distance between two organisms *A* and *B* is defined using the Euclidean Equation:(6)$$D_{AB}=\sqrt{\overset{5}{\underset{i=1}{\sum }}\overset{4}{\underset{j=1}{\sum }}{\left(A_{ij}-B_{ij}\right)}^{2}}$$
where *D_AB_* is the distance between organisms *A* and *B* and *X_ij_* (*X = A* or *B*) is the gene density at the *ij*-th (1 ≤ *I* ≤ 5, 1 ≤ *j* ≤ 4) block. The calculated distance matrix is converted to a phylogenetic tree, as shown in Figure 3B. The RNA viruses are excluded in this tree because both have fewer than 10 proteins and therefore are not statistically significant. 

As expected, the phylogenetic tree clearly separates eukaryotes from prokaryotes. This tree also correctly separates both animals (*H. sapiens* and *Drosophila melanogaster*) and plants (*A. thaliana* and *O*. *sativa*), whereas the basal angiosperm (*A. trichopoda*) and moss (*Physcomitrella*
*patens*) are at the basal locations of the eukaryotic branch. *Mimivirus* is located at the prokaryotic branch and *Pandoravirus* is located at the eukaryotic branch, sitting with the moss *P. patens.* This trend is also expected from the *ΣH:**ΣI* ratios. However, similar to our previous study [37], for all prokaryotes, the phylogeny failed to separate bacteria from archaea without introducing other attributes. Further investigations, such as considering the domain components in the proteins [39], will be required to understand the evolution of the protein entropy. 

The phylogenetic tree presented in Figure 3B (tree of proteomes (ToP)) is by no means for capturing the evolution of species that are usually derived based on multi-sequence alignments (MSAs) of genes. However, this ToP might be visualized as the evolution of structural entropies. It seems that higher complexity favors higher entropic content, especially the entropy-to-information ratios, in the proteins.

## 5. Structural Entropy of Regulatory Proteins

Two categories of proteins—the transcription factors (TFs) and kinases—are further examined from three model organisms, *Escherichia coli*, *A. thaliana*, and *H. sapiens*. Both categories are abundant in the organism and play regulatory roles in the cells. Whilst regulations by the TFs often directly affect the expressions or translations of the genes or transcripts through recognitions and/or interactions with the nucleic acids, the kinases perform post-translational regulations by catalyzing the phosphorylation reactions to the target AAs of proteins (often Ser, Thr, or Tyr) through the adenosine triphosphate (ATP) cofactors.

Figure 4 shows that the eukaryotes (the plant *A. thaliana* and animal *H. sapiens*) possess higher *ΣH:**ΣI* ratios than the prokaryote (*E. coli*), which is consistent with the trend in Figure 2. Moreover, the regulatory mechanisms may influence the *H:I* ratios (*R*’s) of the proteins such that the recognition and interactions with the DNA or RNA targets for the TFs may involve larger amounts of information gains or entropy losses compared to the enzymatic catalysis by the kinases via a lock-and-key-like model. Therefore, we may propose that more intrinsic structural entropies are required for the regulations by TFs than the those required by the kinases, or possibly other enzymes catalyzing different reactions in the cells. These differences are more significant in the eukaryotes (*H. sapiens* and *A. thaliana*) than in the prokaryotes (*E. coli*). Moreover, the *A. thaliana* TFs have a larger *ΣH:**ΣI* ratio compared to those of *H. sapiens*, indicating that the plant TFs may possess overall higher structural entropies than do the animal TFs.

## 6. Discussion

Many algorithms have been developed to predict the probabilities of being disordered for amino acids in proteins, which are presented as the disorder contents of the residues; however, it is not straightforward to quantify how “disordered” a protein is using these approaches. Here, we suggest converting the disorder content to structural entropy, which is continuous and additive and therefore makes it feasible to quantitatively compare two proteins in terms of the degrees of structural entropy and/or information they possess.

As Erwin Schrödinger noted, “life feeds on negative entropy” [40]. The nature of the negative entropy (negentropy) is, actually, the information. Our results suggest that proteins and other biomolecules also feed on information. The capacity of every protein is fixed (Equation (4)), and, therefore, the higher the information content it receives, the lower the entropy it possesses. A recent paper further indicates that the IDPs in the cell are either globally or partially structured. [41] In other words, the structural entropy carried by the IDPs is fully or partially registered by the structural information from the crowded and dynamic intracellular environment of the living cell. 

It might also be expected that the structural entropy/information proposed here has implications in biological processes where structural information-gain/entropy-loss (or the reverse) occur, such as in gene transcriptions, translations, protein folding pathways, protein-protein or protein-ligand interactions, and enzymatic reactions, to name a few. A recent work [42] reported the general transcriptional coactivator *CBP* that binds to two different proteins, the negative feedback regulator *CITED2* (target *T1*) and the hypoxia-inducible factor HIF-1α (target *T2*). The binding affinities of *CBP* to both substrates are very close. However, even at a modest concentration, the substrate *T1* rapidly substitutes *T2* in binding to *CBP*. In addition, it was shown that *T1* exhibits higher disorder contents—and therefore higher structural entropy—than *T2*. Under our approach, the higher specificity of *T1* can be understood as the higher information-gain of *CBP* upon binding to *T1* than that of binding to *T2.*


This approach might be extended to DNA, RNA, and other biomolecules in cases where their entropies and information could be estimated. It should be emphasized here that to quantify the information or entropy, the meaning of the information might be needed. This may especially be the case for the biological information. For instance, the meaning of the information discussed in the present work is the structure of proteins—or, more precisely, the order-to-disorder continuum in the protein world. 

## 7. Methods

Proteomes of organisms from all three domains of life have been surveyed in the present work. Three bacteria include the alphaproteobacterium *Rickettsiales*
*bacterium Ac37b* [43], which may be a close precursor of eukaryotic mitochondria, the cyanobacterium *Synechococcus*
*elongatus PCC 7942* [44], which may represent the precursor of plant plastids, and, recently synthesized, the smallest known free-living organism *JCVI-Syn3.0* [27]. Three archaea include the smallest known free-living archaeon *I. hospitalis*; its parasite, the smallest known archaeon *N. equitans* [45]; and *Lokiarchaeum* (*sp. GC14_75*) [30]. The latter was reported to possess eukaryotic genes, such that its close relative, may have served as the host to accommodate an obligate parasitic bacterium, and the archaeon and bacterium accidently merged to produce the first eukaryotic cell some 2 billion years ago. The eukaryotic organisms include three plant species: the basal angiosperm *Amborella*
*trichopoda* [46], the monocot plant *O. sativa* [29] (rice), and the eudicot and model plant *A. thaliana* [47]; two animal species: *H. sapiens* [48,49] (human) and *D. melanogaster* [50] (fruit fly); the moss *P. patens* [51]; and the yeast *S. cerevisiae* [52]. Several viral proteomes were also studied, including two RNA viruses with less than 10 genes from the *Ebola* virus and human coronavirus (which causes the common cold), plus two giant DNA viruses (giruses), the *Mimivirus* [53], and *Pandoravirus* (or *P. salinus*) [54]. In addition, protein sequences from the database of experimentally confirmed disordered proteins, DisProt (v7.0) [25], and those from the protein data bank [26] (PDB, up to December 16, 2016) are also analyzed. The redundant sequences from the PDB have been removed, and each entry represents a unique protein sequence in the analysis: For each of the protein sets, if there are multiple identical copies of a sequence, such as those from identical chains from a homo-dimer or other oligomers, only one unique sequence is retained for further analysis.

The PONDR-VSL2 algorithm [55] was applied to predict the ID content of all residues in a protein. The obtained disorder content was then converted to probabilities that associated with the structural entropy, using Equation 3. A neighbor-joining method, from the T-REX web server [56], was used to convert the distance matrix to phylogenetic trees. More details on converting the disorder contents to structural entropy, and the fitting of the structural capacities using different models, can be found in the Appendix A. 

GitHub repository: https://github.com/haoboguo/Protein-Structural-Entropy. Code written in *R* (*random.seq.R*) was used to generate 500 random sequences with random capacities in the range [50, 800]. A shell-script was written (*entropy.csh*) to perform protein structural entropy/information/capacity evaluations, using Equation (3). The PONDR-VSL2 predictor was downloaded from http://www.dabi.temple.edu/disprot/predictor.php. JAVA is required to run this code to perform disorder predictions. See descriptions in above repository.

## Figures and Tables

**Figure 1 entropy-21-00591-f001:**
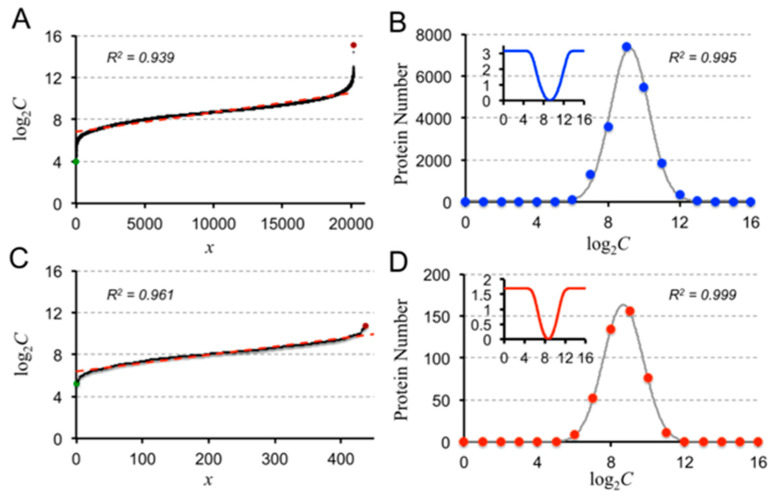
Distribution of the structural capacity *C* (left) and Gaussian distributions of log_2_*C* (right) in proteomes of (**A**,**B**). *Homo sapiens* (blue) and (**C**,**D**) *JCVI-Syn3.0* (red). In the left figures, the brown and green dots are the longest and shortest proteins of the proteomes, respectively. The red-dashed lines are the exponential fittings. The insets in **B** and **D** convert the probability distributions of proteins to the potential of mean forces (in kcal/mol) at 300 K using Equation (5). Distributions of *C* using different fitting models are summarized in Appendix A.

**Figure 2 entropy-21-00591-f002:**
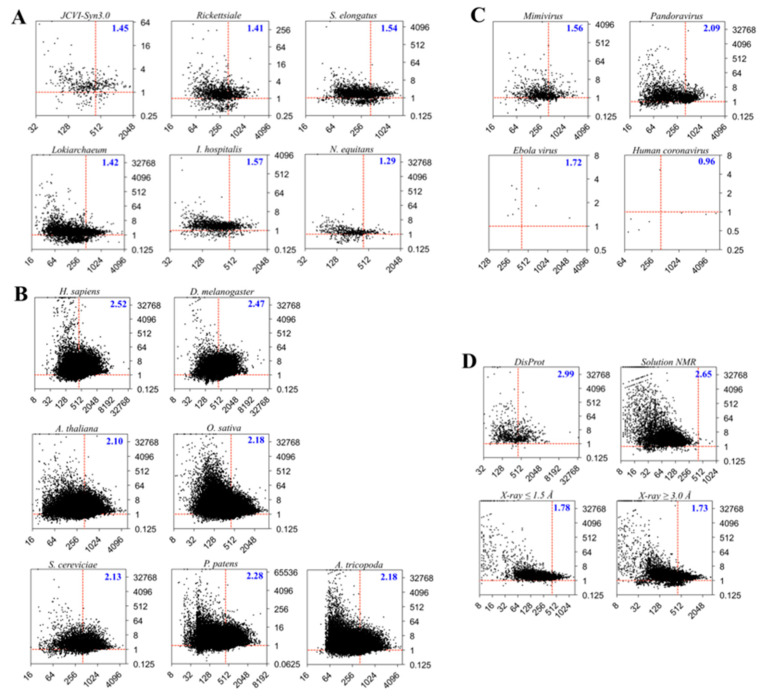
Distributions of proteins in the *CR-*space from proteomes of (**A**) prokaryotes, (**B**) eukaryotes (**C**) viruses, and (**D**) protein sets from DisProt and PDB [25,26]. Each protein *i* is represented by a black dot. The horizontal axis is *C_i_* = *H_i_* + *I_i_* and the vertical axis is the ratio between *H_i_* and *I_i_* such that *R_i_*
*= H_i_/I_i_*. The horizontal dashed line (red) corresponds to *R_i_* = 1; proteins above this line are entropy-rich with *H_i_* > *I_i_*, and those under this line are information-rich with *H_i_* < *I_i_*. The vertical dashed line serves as a reference, using the median capacity of the human proteome at *C_i_* = 417. The total *H*-to-*I* ratios (Σ*H:*Σ*I*) are given in the maps. Note that the boundaries of the horizontal (*R*) and vertical (*C*) axes vary across different proteomes and/or protein sets.

**Figure 3 entropy-21-00591-f003:**
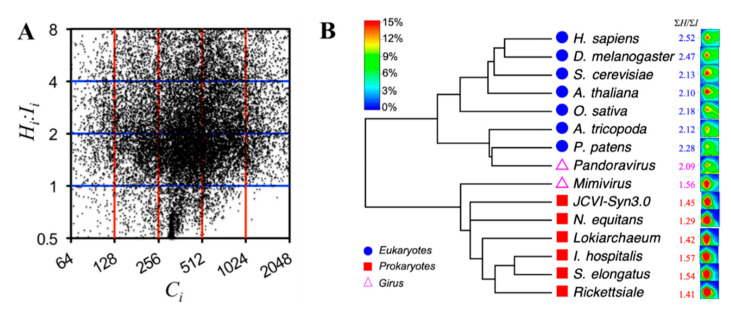
(**A**) Proteins distributed in the *CR*-space that is divided into 20 blocks exemplified by the proteome of *H. sapiens*. Only proteins with *C_i_* between 64 and 2048 and *H_i_:I_i_* between 0.5 and 8 are shown; other proteins are omitted for clarity. (**B**) Phylogenetic tree reconstructed using the Euclidean distances calculated from the gene densities at different blocks. The divergent history may not be covered by the branch lengths that are estimated from the gene densities in the *CR*-space. The prokaryotes (both bacteria and archaea) are shown in red squares, the eukaryotes in blue spheres, and the giant DNA viruses in purple hollow triangles. The contour maps show the densities of proteins (color bar at top left) distributed in the *CR*-space.

**Figure 4 entropy-21-00591-f004:**
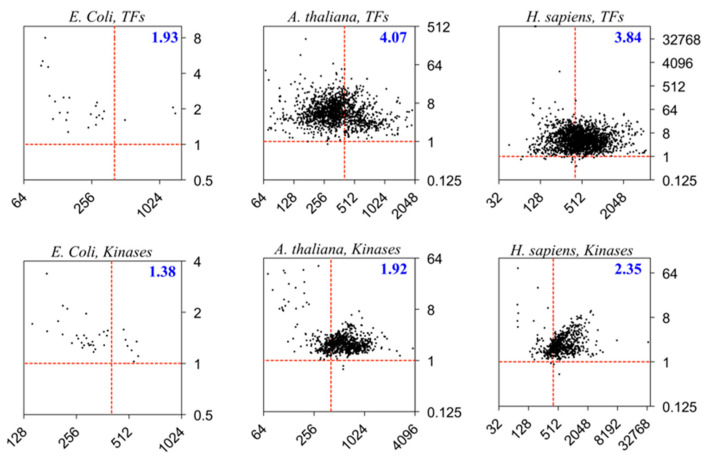
Distributions of proteins in the *CR*-space of transcription factors (top panel) and kinases (bottom panel) in the three model organisms *E. coli* (left), *A. thaliana* (center), and *H. sapiens* (right). The proteome of *E. coli* has 25 transcription factors (TFs) and 35 kinases, the proteome of *A. thaliana* has 1508 TFs and 899 kinases, and the proteome of *H. sapiens* has 2017 TFs and 650 kinases. All protein sequences are obtained from the UniProt. As in Figure 2, each protein is represented by a black dot in the maps, the horizontal axis is for the structural capacity *C_i_*_,_ and the vertical axis is for the *H:I* ratios *R_i_*. Σ*H:*Σ*I* is given for each group of proteins.

**Table 1 entropy-21-00591-t001:** Summary of proteomes and datasets in the present study.

	Species	*C* _max_	*C* _min_	*R* _max_	*R* _min_	N	Σ*H*/Σ*I*
**Prokaryotes**	*JCVI-Syn3.0*	(1789, 1.43)	(37, 56.72)	(63, 66.96)	(224, 0.35)	438	1.451
	*Rickettsiale*	(2243, 2.02)	(31, 6.64)	(55, 397.55)	(170, 0.28)	1780	1.413
	*S. elongatus*	(1807, 1.31)	(29, 0.94); (29, 2.88)	(54, 7713.3)	(62, 0.38)	2612	1.538
	*Lokiarchaeum*	(3592, 1.65)	(20, 3.90); (20, 53.05)	(22–47, ∞), 3 proteins	(99, 0.23)	5384	1.423
	*I. hospitalis*	(1392, 2.31)	(33, 089)	(52, 2887.9)	(219, 0.33)	1431	1.568
	*N. equitans*	(1390, 0.94)	(45, 5.58)	(55, ∞)	(160, 0.25)	540	1.290
**Eukaryotes**	*H. sapiens*	(34350, 2.52)	(16, 4.17)	(25–288, ∞), 37 proteins	(304, 0.35)	20193	2.525
	*D. melanogaster*	(22949, 9.42)	(11, 11.67); (11, 66.48)	(25–118, ∞), 9 proteins	(139, 0.23)	13700	2.469
	*S. cerevisiae*	(4910, 1.78)	(16, 1.37)	(25, ∞), 2 proteins	(88, 0.30)	5917	2.127
	*A. thaliana*	(5393, 1.72)	(16, 4.03)	(19–81, ∞), 10 proteins	(23, 0.25)	26396	2.096
	*O. sativa*	(4957, 1.64)	(9, 249.0)	(10–142, ∞), 12 proteins	(33, 0.28)	48782	2.180
	*A. trichopoda*	(4988, 1.72)	(29, 0.78)	(53–227, ∞), 14 proteins	(73, 0.33)	26461	2.124
	*P. patens*	(5199, 1.75)	(13, 12999)	(20–131, ∞), 13 proteins	(102, 0.10)	32400	2.278
**Viruses**	*Mimivirus*	(2959, 1.66)	(25, 0.56)	(282, 5319.8)	(38, 0.18)	979	1.561
	*Pandoravirus*	(2321, 1.86)	(26, 8.69); (26, 57.54)	(29, ∞)	(29, 0.28)	2541	2.092
	*Ebola virus*	(2212, 1.27)	(251, 1.36)	(288, 3.28)	(2212, 1.27)	9	1.724
	*Human coronavirus*	(6758, 0.95)	(77, 0.48)	(389, 4.68)	(77, 0.48)	8	0.960
**Datasets**	*DisProt*	(34350, 2.52)	(33, 21.62)	(72–107, ∞), 3 proteins	(256, 0.49)	803	2.992
	PDB, Solution NMR	(828, 2.09)	(9, 4.07–∞), 28 proteins	(9–53, ∞), 169 proteins	(15, 0.31)	8476	2.646
	PDB, X-ray, ≤1.5 Å	(1305, 1.63)	(9, 0.65–∞), 31 proteins	(9–52, ∞), 60 proteins	(29, 0.47)	5842	1.779
	PDB, X-ray, ≥3.0 Å	(3450, 1.59)	(9, 0.68–∞), 29 proteins	(9–40, ∞), 66 proteins	(115, 0.32)	10636	1.734

All data are represented using (*C*, *R*), where *C* is the structural capacity (vertical axis in Figures 2 and 4) and R is the *H*-to-*I* ratio (horizontal axis in Figures 2 and 4), respectively. The data for proteins with extreme *C*’s or *R*’s are given. For those with multiple entries, a range for *C* or *R* and the total number of proteins is listed. The infinite (∞) values in *R* come from the totally disordered proteins with *I* = 0 and *H = C*, respectively. N represents the total number of different proteins in the proteome or dataset; for eukaryotic proteomes, only the primary protein at each gene locus is counted. The last column gives the total *H* divided by total *I* of all proteins in the proteome or dataset. No average *R*’s are given because there are proteins with *R* of ∞ in all eukaryotes, datasets, and the archaea *Lokiarchaeum* and *Nanoarchaeum*
*equitans*. Details of the names of organisms and datasets can be found in the Methods section.

**Table 2 entropy-21-00591-t002:** Intervals that partition the *CR-*space into 5 × 4 blocks.

***Capacity***	1	2	3	4	5
***C_i_***	[1,127]	[128,255]	[256,511]	[512,1023]	[1024, ∞)
***Ratio***	1	2	3	4	
***R_i_***	[0,1)	[1,2)	[2,4)	[4,∞)

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
