# Peer review of "A Suggestion of Converting Protein Intrinsic Disorder to Structural Entropy Using Shannon’s Information Theory"

_entropy, 2019, doi:10.3390/e21060591_

Round 1

Reviewer 1 Report

The authors applied the Shannon’s entropy theory to contents of IDRs, and compared the proteomes of some of the model organisms based on this analysis. However, I think the results seem to be another format of the well-known results previously reported, and I could not find something new in this analysis. The authors must present benefits of employing this method. In addition, the following points must be revised.

1)    The methods are not well described. I think that the authors need to know disordered/ordered for each of the residues. However, I could not find how to know this information. In addition, if you used some of the prediction programs, you should conduct same analysis by other prediction programs, because differences between the prediction methods must make big impacts on the results.

2)    Figure 1 is not understandable. What is the red dashed lines in A and C ? I could not understand the label “Protein Number”. Then, I could not see what the vertical axis is. Also, what is the horizontal axis in A and C. It has the labels “x”, however, you used small x in the Eq.3 (though it has suffix i). Xi in Eq.3 is a probability, then it must be less than 1, but the horizontal axis ranges from 0 to 20000. This confused me very much.

3)    I could not understand “protein distribution” in line 131.

4)    Line 137, “the protein density with ni the protein numbers…” may be typo. Something may be needed between “ni” and “the protein”. This kind of careless builds frustration.

5)    In Table 1, the organism names are presented in common names, scientific names, and (maybe) strain names (JCVI-Syn3.0).

6)    In Table 1, the authors used PDB datasets. I think that a factor of this analysis is a length of a protein. However, the chains deposited in PDB are truncated in order to determine their 3D structures in many cases. Then, the chain lengths in PDB contain many artifacts. I wonder if PDB is suitable for the comparison of the proteomes here.

7)    The scale of the axes is better to be the same. If this is difficult, something is needed to present these results reflecting the scale differences.

Author Response

Reviewer 1 comments

The authors applied the Shannon’s entropy theory to contents of IDRs, and compared the proteomes of some of the model organisms based on this analysis. However, I think the results seem to be another format of the well-known results previously reported, and I could not find something new in this analysis. The authors must present benefits of employing this method. In addition, the following points must be revised.

Reply: Thank you very much for this comment. We do not agree with the point that our results are just “another format of the well-known results previously reported”. In this revision a paragraph was added to the Introduction section:

“A fundamental difference between D and H is that the structural entropy H is an additive quantity, whereas the ID score D is not (see the Appendix in the supporting material). That is, adding up structural entropies of all residues leads to the structural entropy of the entire protein. Whereas we cannot infer the ID score of a protein by adding up the ID scores of all its residues. We suggest that the (intrinsic) structural entropy either of a single residue in the protein or of the entire protein can be directly calculated from a disorder predictor. Therefore, our suggestion indicates that it is feasible to quantitatively compare structural entropies of two proteins, regardless of their respective disorder percentages, which are rough and qualitative measures of the protein disorder.”

More details can be found in the Appendix. According to the review comments we feel it is necessary to put the potential benefits of employing our method in the Introduction. Indeed, using the entropy suggestion we will be able to quantitatively compare different proteins in terms of order/disorder, and can avoid the approximate or analogous descriptions such as 50% disordered residues.

1)    The methods are not well described. I think that the authors need to know disordered/ordered for each of the residues. However, I could not find how to know this information. In addition, if you used some of the prediction programs, you should conduct same analysis by other prediction programs, because differences between the prediction methods must make big impacts on the results.

Reply: Thanks for this suggestion! We have revised the methodology section. In addition, we have put all scripts in calculating H and I in a GitHub repository (Method section). We agree that the prediction methods may have big impacts on the results. Nevertheless, the focus of our manuscript is not comparing different protein disorder predictors (so far there are 60+ different predictors). Rather, our aim is to provide a suggestion to convert the disorder contents into structural entropies.

2)    Figure 1 is not understandable. What is the red dashed lines in A and C ? I could not understand the label “Protein Number”. Then, I could not see what the vertical axis is. Also, what is the horizontal axis in A and C. It has the labels “x”, however, you used small x in the Eq.3 (though it has suffix i). Xi in Eq.3 is a probability, then it must be less than 1, but the horizontal axis ranges from 0 to 20000. This confused me very much.

Reply: Sorry for having not clearly written the Figure caption. We have added the corrections: the red-dashed lines in A and C are the exponential fitting to the capacities (protein lengths). Log2-scale had been used in the y-axis such that the fittings appear as linear lines. The parameters for the fittings can be found in Table S1 of the supporting materials.

We are sorry for the confusion that you experienced! Also thank you very much for pointing it out! In the new version we changed the title of x-axis as “Protein Serial Number”, for which all proteins of the proteome (or data set) have been ranked hierarchically by capacities (or protein lengths). For example, for n proteins, the protein with serial number 1 is the shortest whereas that with serial number n is the longest. This serial numbers are not that used in eq.3. Yes you are absolutely correct that xi(i is for the i-th residue) in eq. 3 must be less than 1; in fact, xishould be in range of [0, 0.5]. xi= 0.5 indicates that the residue has the highest entropy of 1 (or fully disordered), in accordance to Shannon’s equation. 

Here we hope to clarify that in this manuscript we used a capital X for a certain protein and a lower case xi(with suffix i) for the i-th residue of this protein. Based on this comment we also corrected a typo in the main text: in line 98 of the original version, it should be “0 £H(X) [or I(X)] £C”, where X is a particular protein and C is its capacity.

3)    I could not understand “protein distribution” in line 131.

Reply: Thank you catching this point! We changed it to “protein-capacity distribution”, which is equivalent to “protein-length distribution” as in Ref. 24 (Zhang 2000).

4)    Line 137, “the protein density with ni the protein numbers…” may be typo. Something may be needed between “ni” and “the protein”. This kind of careless builds frustration.

Reply: We are sorry for the confusion! We have changed this sentence to “riis the density of proteins at the i-th histogram of protein-capacities”

5)    In Table 1, the organism names are presented in common names, scientific names, and (maybe) strain names (JCVI-Syn3.0).

Reply: Thanks for point this out!The details of the organism names can be found in the Methods section. We added this clarification in the table legend.

6)    In Table 1, the authors used PDB datasets. I think that a factor of this analysis is a length of a protein. However, the chains deposited in PDB are truncated in order to determine their 3D structures in many cases. Then, the chain lengths in PDB contain many artifacts. I wonder if PDB is suitable for the comparison of the proteomes here.

Reply:Thank you for this suggestion! We are aware that many proteins deposited in the PDB have truncated structures. And in many cases only a particular domain had been reported. In most cases, however, the sequences downloaded from the PDB are the full-length proteins or domains. The residues without coordinates are summarized in the PDB file as missing residues (“M RES” in remarks). Our goal here is not to compare the PDB structures (sequences) with the proteomes; rather, not surprisingly, we found that the X-ray structures generally have lower structural entropies compared with the NMR structures, and reported resolutions has a negligible impact to the structural entropies.

7)    The scale of the axes is better to be the same. If this is difficult, something is needed to present these results reflecting the scale differences.

Reply: Thanks for point this out! We used log-scales to represent the distribution of proteins in the C-R space (Figure 2) and that the scales are determined by the upper and lower borders of the C’s (capacities) and R’s (H:I). We added this clarification in the Figure legend. Also note that we have changed the name of HI-space to CR-space as the latter should be a more appropriate name, thanks to the comments from another reviewer.

Reviewer 2 Report

In this paper titled “A Suggestion of Converting Protein Intrinsic Disorder to Structural Entropy using Shannon’s Information Theory”, the authors introduced the application of Shannon’s information theory to measure the intrinsic disorder content in terms of structural entropy. Initially, they mentioned the mathematical expression for Structural entropy (H), structural information (I) and structural capacity (C). Then the intrinsic disorder content of the proteins in different proteomes ranging from prokaryotes to eukaryotes was estimated in terms of the H, I and C. They also proposed the application of the concept to show the pattern of structural entropies and information of the proteins over the phylogenic tree and for the transcription factors, kinases as well. Although authors approached a good way to convert the intrinsic disorder to structural entropy more efficiently, they lacked a more detailed explanation to clarify the whole analysis as they were introducing a new concept in estimating structural entropy. In my opinion, there are few suggestions which would improve the manuscript more acceptable to the broader audience and to the journal.

1)      In the manuscript, authors used “negative entropy” for several times, what would be the physical explanation of it?

2)      The introduction of structural entropies and information theory were not explained properly. I think that more explanation of these apparently new terms to determine the disorder is required.

3)      I am wondering if authors would able to put more insightful discussion of structural entropies and information theory in the background of IDPs and it would be helpful if authors interpreted these terms with a simple random sequence so that a non-mathematical person can also understand.

4)      What is the difference between entropy-rich and information-rich in their analysis? More clarification in this direction would be more insightful.

5)       For the human proteome, the well-depth is 3.16kcal/mol centered at C=592 whereas JCVI-syn3.0 has 1.70 kcal/mol centered at 401. What would be the interpretation of these observations?

6)      In the evolution of protein structural entropies and information part, the explanation is very limited and I think that interpretation with mathematical details makes this paper more interesting.

7)      In Figure 3B, the contour diagrams are not visible. Enlarged one will be helpful.

Author Response

Reviewer 2 comments:

In this paper titled “A Suggestion of Converting Protein Intrinsic Disorder to Structural Entropy using Shannon’s Information Theory”, the authors introduced the application of Shannon’s information theory to measure the intrinsic disorder content in terms of structural entropy. Initially, they mentioned the mathematical expression for Structural entropy (H), structural information (I) and structural capacity (C). Then the intrinsic disorder content of the proteins in different proteomes ranging from prokaryotes to eukaryotes was estimated in terms of the H, I and C. They also proposed the application of the concept to show the pattern of structural entropies and information of the proteins over the phylogenic tree and for the transcription factors, kinases as well. Although authors approached a good way to convert the intrinsic disorder to structural entropy more efficiently, they lacked a more detailed explanation to clarify the whole analysis as they were introducing a new concept in estimating structural entropy. In my opinion, there are few suggestions which would improve the manuscript more acceptable to the broader audience and to the journal.

Reply: Thank you for this positive comment. We added a paragraph in the Introduction indicating why our suggestion would be useful. Especially that our approach is in line with Shannon’s original proposal, in which the entropy H is continuous, monotonic increasing and, importantly, additive. The protein intrinsic disorder content (D) may satisfy the first two criteria, however, it is not additive and therefore one cannot sum up the disorder contents of all residues in a protein and specify that the protein has a disorder of  SDi. However, using our approach it is appropriate to say that a protein has the structural entropy of  SHi, where Hiis the entropy of the i-th residue.

1)      In the manuscript, authors used “negative entropy” for several times, what would be the physical explanation of it?

Reply: Thank you for noticing the negentropy (negative entropy) appeared in our MS. We added citations of L. Brillouin (Refs. 22 and 23) who coined this term (shortly after Shannon, and cited Schrodinger, of course). 

The physical explanation of negentropy had been largely discussed in the literature and well-understood from the Landauer’s principle (1961) that erasure of information (i.e., negative entropy) inherently dissipates heat (into the environment). However, we feel that it might be very difficult for our work to join in this discussion --- although there might be energetic equivalence to the structural entropy (i.e., from binding, folding etc.). 

2)      The introduction of structural entropies and information theory were not explained properly. I think that more explanation of these apparently new terms to determine the disorder is required.

Reply: Thank you for this suggestion! We hope that the new paragraph we added into the introduction can help to explain why the structural entropy is better than simply interpret the disorder contents of residues.

3)      I am wondering if authors would able to put more insightful discussion of structural entropies and information theory in the background of IDPs and it would be helpful if authors interpreted these terms with a simple random sequence so that a non-mathematical person can also understand.

Reply: Thank you very much for this great suggestion!

We have generated 500 random sequences with the capacities randomly chosen from [50, 800]. Interestingly, the SH:SI ratio is 1.020 for this set. This is smaller than all cellular organisms, but is close to that of human coronavirus. This finding suggests that the larger H compared to I for both eukaryotes and prokaryotes may not be random. We added a short sub-section in Results. The R-code and for construction the random sets, the random set that we generated and results have been put into a GitHub repository.

4)      What is the difference between entropy-rich and information-rich in their analysis? More clarification in this direction would be more insightful.

Reply: We used the horizontal border line at R = 1, to separate entropy-rich from information-rich. It is especially clear from the random-sequence analysis (thank you again for your great suggestion). It is clear that the random-sequences has a balance ratio of R »1; whereas all eukaryotic and prokaryotic proteomes are entropy-rich. It seems the more complex organism has larger SH:SI ratios. These fact written in the MS all help to clarify that there is a R = 1 boundary between entropy-rich and information-rich.

5)       For the human proteome, the well-depth is 3.16kcal/mol centered at C=592 whereas JCVI-syn3.0 has 1.70 kcal/mol centered at 401. What would be the interpretation of these observations?

Reply: We added this discussion because previously people found the protein length (i.e., capacities) have exponential distributions (e.g., Zhang 2000). To our knowledge, this may be the first time to report that there is a hidden Gaussian distribution here. Unfortunately, our current understanding could not specify deeper interpretation of these observations. We hope to say this might be driven by the complexity of organisms, but lack evidences. We wish to disclose the observation here, waiting for the insights from other experts.

6)      In the evolution of protein structural entropies and information part, the explanation is very limited and I think that interpretation with mathematical details makes this paper more interesting.

Reply: Thanks for this suggestion! We have summarized the mathematical details for calculating the distances between organisms and reconstructing the phylogeny. This is a very naive, somewhat intuitive approach. We have cited the recent work by Yu et al. (ref. 38) which was published after our manuscripts had been finished. Their approach can even distinguish Archaea from Bacteria! Whereas our approach cannot (although we did see correct patterns in the eukaryotic branch). That is also the reason we specified that “Further investigations, such as considering the domain components in the proteins38, will be required to understand the evolution of the protein entropy.”

7)      In Figure 3B, the contour diagrams are not visible. Enlarged one will be helpful.

Reply: Well, thanks for this suggestion! In another work we have summarized similar contour diagrams in SM (Guo et al. 2018). Here we intend to present a broader view of how the eukaryotic and prokaryotic organisms are separated based on the protein distributions in the CR-space. We feel that the current version should be good.

Reviewer 3 Report

 Please discuss in more detail what are the advantages of the presented theory, when compared to the usual description of protein disorder. What kind of applications might it have? At the end of the introduction the reviewer was still hoping that the method might help to judge the simple question if a protein is ordered or disordered with 50% disordered residues, but this question remained unanswered. In Discussion the example of T1/T2 CBP binding is mentioned, that the higher specificity of T1 can be understood as higher information-gain. Substrate binding is governed by the (Gibbs) free enthalpy of binding which has enthalpic an also entropic contributions. In this specific case and generally in the whole manuscript matching of the thermodynamic and information theory enthalpy could be emphasized.

 Please give more details about results presented in Figure 2D. According to the reviewers understanding proteins with higher disorder are “entropy rich” and should be shifted to the top of the figure with higher R values. The total H/I ratio is higher for DisProt proteins than for the PDB datasets, but the figures do not reproduce the fact that DisProt proteins are more disordered or entropy rich. Maybe the H-I (or better R-C) representation is not the best choice. Could be there another representation which could visually separate ordered and disordered structures?

The methods should be explained detailed. How was redundancy removed? Which DisProt version (v0.5?) was used? Complete protein sequences or only disordered segments were used? The former is presumed but it is not explicitly written in the manuscript. The reviewer feels that the case of Titin is not adequately discussed, either. The fact that 60 (+66) proteins from the PDB database with high resolution X-ray structures are fully disordered is quite surprising. This should be discussed, the list of these proteins could be added to the Supplement. These proteins might overlap with the recently published MFIB database, which contains oligomeric proteins disordered in monomeric but ordered in oligomeric form. Thus the oligomeric structures can be found in the PDB database. In the Datasets paragraph 8 DisProt proteins are mentioned which have an R value below 1. The list of these proteins could also be added to SM and could also be discussed in more detail. Size of the proteins, and disordered segments might also contribute to this.

In Introduction the PONDR family is mentioned as a representative set of prediction algorithms, which is true. The VSL2 method is one (if not the best) single prediction method.  However there are other methods which do not make use of neural networks, furthermore there are decision based consensus methods which are even more accurate in some aspects. What changes could the use of another disorder prediction method induce? Could experimentally validated disorder information be included into the method?

Further minor findings:

Shouldn’t the references of DisProt and PDB database included at the first occurrence?

In the case of the Shannon approach x(i)=d(i)/2. Is the value range of x [0,0.5]?

The “titin” word is missing from the “Supplementary materials include” chapters.

In Supplementary Materials S3A is written several times instead of S2A and S4 instead of S3.

In the case of transcription factors in Fig4 the total H/I ratio is higher for the A.thaliana case. This is a somewhat surprising result and could be discussed

Author Response

Reviewer 3 comments:

Please discuss in more detail what are the advantages of the presented theory, when compared to the usual description of protein disorder. What kind of applications might it have? At the end of the introduction the reviewer was still hoping that the method might help to judge the simple question if a protein is ordered or disordered with 50% disordered residues, but this question remained unanswered. 

Reply: Thank you very much for the comments! We are really sorry that for having not address the question we asked at the beginning of the manuscript. In the original manuscript we had put deductions into the Appendix in the supporting materials without a generalized conclusion appearing in the main text. In the new version, a new paragraph was added in the Introduction:

“A fundamental difference between D and H is that the structural entropy H is an additive quantity, whereas the ID score D is not (see the Appendix in the supporting material). That is, adding up structural entropies of all residues leads to the structural entropy of the entire protein. Whereas we cannot infer the ID score of a protein by adding up the ID scores of all its residues. We suggest that the (intrinsic) structural entropy either of a single residue in the protein or of the entire protein can be directly calculated from a disorder predictor. Therefore, our suggestion indicates that it is feasible to quantitatively compare structural entropies of two proteins, regardless of their respective disorder percentages, which are rough and qualitative measures of the protein disorder.”

We hope this statement could answer the question. If “50% disordered residues” is an analogous description, the structural entropy H of a protein would be accurate owing to that the structural entropy is additive (just as its counterpart of the thermodynamic entropy).

In Discussion the example of T1/T2 CBP binding is mentioned, that the higher specificity of T1 can be understood as higher information-gain. Substrate binding is governed by the (Gibbs) free enthalpy of binding which has enthalpic an also entropic contributions. In this specific case and generally in the whole manuscript matching of the thermodynamic and information theory enthalpy could be emphasized.

Reply: The information entropy does have the thermodynamic meaning, e.g., governed by Laudauer’s principle erasure of 1 bit of information is associated to dissipation of 1 kBT of energy to the environment (where kBis the Boltzmann’s constant and T the temperature). However, we would not insist that we can convert the (structural) information-gain or entropy-loss to the thermodynamic entropy or free energy. Although we suspect that Schrodinger might actually have free energy concept in his mind when he stated that “life feeds on negative entropy”. In addition, our approach set the total number of residues as its capacity of a protein, similar to the “bit” (binary digit) defined by Shannon. Shannon’s equation has a positive constant K, meaning that we might use different unit to describe the information entropy depending on the logarithmic function chosen. As mentioned in the Appendix, we might even use a three-state-system to estimate the structural entropy of a protein. But none of these approaches can catch up how many kcal/mol for the binding between T1 or T2 to CBP. We only propose a hypothesis here in the Discussion, because there could be many possibilities for the suggested “higher-information gain”, such as coupled folding and binding or equilibrium shift. We have added a paragraph in Discussion to address the coupled folding and binding in IDPs. For the CBP example, we hope to keep it as is.

Please give more details about results presented in Figure 2D. According to the reviewers understanding proteins with higher disorder are “entropy rich” and should be shifted to the top of the figure with higher R values. The total H/I ratio is higher for DisProt proteins than for the PDB datasets, but the figures do not reproduce the fact that DisProt proteins are more disordered or entropy rich. Maybe the H-I (or better R-C) representation is not the best choice. Could be there another representation which could visually separate ordered and disordered structures?

Reply: Thanks for this suggestion! We have changed the HI-space as CR-space throughout the manuscript, which is more appropriate. We agree that using the total H/I ratio alone is not sufficient to distinguish different proteomes or protein sets; first, the number of proteins vary among different figures in Figure 2D (and other proteomes), second, we used SHi/SIi, hence the proteins with higher capacities have higher weights to the total H/I ratio: for example, if there are two proteins, whose (entropy, information) pairs are A(9,1) and B(100,900), also one has H/I ratio of 9 (protein A) and 1/9 (protein B), the total ratio would be 109/901, close to that of B.

Also because of this reason, to compare different proteomes, we have used a Euclidean equation (e.q. 6) to calculate the “distances” between different proteomes, which resulted to the tree shown in Fig. 3B. To our knowledge, this tree not only separate the eukaryotes form prokaryotes, but also set a roughly “correct” order for plants, yeast and animals; even moss (P. patens) and the basal angiosperm (A. tricopoda) sit near the root of the eukaryotic branch, and that the monocot (O. sativa) appears earlier than the dicot (A. thaliana), and so on.

The methods should be explained detailed. How was redundancy removed? Which DisProt version (v0.5?) was used? Complete protein sequences or only disordered segments were used? The former is presumed but it is not explicitly written in the manuscript. The reviewer feels that the case of Titin is not adequately discussed, either. The fact that 60 (+66) proteins from the PDB database with high resolution X-ray structures are fully disordered is quite surprising. This should be discussed, the list of these proteins could be added to the Supplement. These proteins might overlap with the recently published MFIB database, which contains oligomeric proteins disordered in monomeric but ordered in oligomeric form. Thus the oligomeric structures can be found in the PDB database. In the Datasets paragraph 8 DisProt proteins are mentioned which have an R value below 1. The list of these proteins could also be added to SM and could also be discussed in more detail. Size of the proteins, and disordered segments might also contribute to this.

Reply: Thanks for this comment. We used DisProt v7.0, corrected in the manuscript. We have added the information-rich proteins from DisProt v7.0 into the supplementary materials. We also modified the description for the Titin protein. The fully-disordered proteins from both X-ray 1.5 and X-ray 3.0 sets and with capacity larger than 20, have been summarized in Tables S2 and S3 in the supplementary mateirals. We added discussions to these proteins and compared our list with the MFIB database, but none of the proteins in Tables S2 and S3 have been collected in the MTIB database.

In Introduction the PONDR family is mentioned as a representative set of prediction algorithms, which is true. The VSL2 method is one (if not the best) single prediction method.  However there are other methods which do not make use of neural networks, furthermore there are decision based consensus methods which are even more accurate in some aspects. What changes could the use of another disorder prediction method induce? Could experimentally validated disorder information be included into the method?

Reply:Thank you for this comment! We agree that the prediction methods may change the predicted results. VSL2 may not the best predictor. Furthermore, it cannot predict the selenoproteins found in the human proteome. However, this is the only predictor that we found we could obtain and compile by ourselves, and that its output data structure is easy to be incorporated into scripts for large protein sequences. Moreover, the focus of our manuscript is not comparing different protein disorder predictors (so far there are 60+ different predictors). The aim of our manuscript is to provide a suggestion to convert the disorder contents into structural entropies. 

Further minor findings:

Shouldn’t the references of DisProt and PDB database included at the first occurrence?

Reply: Thank you for the suggestion. It had been corrected.

In the case of the Shannon approach x(i)=d(i)/2. Is the value range of x [0,0.5]?

Reply: Yes, you are correct that xiis in range of [0,0.5]. As mentioned in the Appendix, “disordered” or “ordered” should not be considered as the two states of a protein. We can assume a two-state-system and “disordered” or “ordered” is the probability that the protein state is already determined (by the sequence) or not. As mentioned earlier and in the example shown in Fig.S2, we could assume a 3-state-system (for which each residue contributes log23 = 1.585), and so on, the difference would be like the positive constant in Shannon’s equation.

The “titin” word is missing from the “Supplementary materials include” chapters.

Reply: Thank you! We have corrected it.

In Supplementary Materials S3A is written several times instead of S2A and S4 instead of S3.

Reply: Thank you! We have corrected the errors.

In the case of transcription factors in Fig4 the total H/I ratio is higher for the A.thaliana case. This is a somewhat surprising result and could be discussed

Reply: Thank you for this suggestion! Yes, we observed that A. thaliana has a higher SH:SI ratio than H. sapiens (4.07 vs 3.84). We have added a sentence in the MS to address this finding. But we could not propose more interpretations without deeper investigations.

Round 2

Reviewer 1 Report

This version is fine. I have some comments on the lines from 249 to 261. Table S2 and S3 have many collagens. Collagen is a structural protein, which has the repeat sequences to form collagen helix. Generally, a biased amino acid composition tends to be predicted as disordered. Then, I wonder these examples were prediction errors. However, these lists contain some examples with the evidences of order/disorder transition. The database, IDEAL, has collected IDR regions showing order/disorder transition, called ProS in this database. 3u85:B, 4fq3:B, and 4gu0:E are found in IDEAL as ProSs in the entries of IID00379, IID00451, and IID00239. The references of these examples to IDEAL must add value to this paper.

Author Response

Reply: Thank you so much for the positive comments! We agree that it might be prediction errors of the PONDR predictor for the collagen sequences in PDB. However, this would not affect the suggestion of our manuscript of substituting the intrinsic disorder to structural entropy. We have mentioned this point explicitly in the revised manuscript: “It may be possible that the PONDR predictor did not perform well for these sequences, particularly the collagen sequences (Tables S2 and S3)”. We have cited the IDEAL database, and the sequences in Tables S2 and S3 that overlap with IDEAL have been labeled. Thanks for this suggestion!

Reviewer 2 Report

No Comments. Best of luck.

Author Response

Reply: Thank you!

Reviewer 3 Report

 The manuscript was improved, however there are a couple of remarks.

 Methods chapter feels incomplete, for example redundancy removal is still not defined.

 The readability of the text on the figures should be improved.

 Is the reference numbering correct? Especially are 25 and 26 correct for PDB and Disprot?

 The disorder prediction of the monomeric high resolution structures is probably an artifact, other methods do not predict them disordered. The other structures are homo trimeric collagen (and a zipper) structures, it is not surprising that they are predicted as disordered on their own.

 The paragraph comparing D and H in the Introduction is welcome, but the question was rather "asking" for a paragraph in Dicussion about general usability of the method.

Author Response

The manuscript was improved, however there are a couple of remarks. Methods chapter feels incomplete, for example redundancy removal is still not defined.

Reply: In our previous revision the redundancy removal had been added in the Results/Dataset section. In this revision we have added it to the Methods section.

The readability of the text on the figures should be improved.

Reply: The figures may be enlarged a little, and we will also provide figures in higher resolutions. Thanks for pointing it out.

 Is the reference numbering correct? Especially are 25 and 26 correct for PDB and Disprot?

Reply: In this revision, we have double checked all references. In addition, the reference style has been changed using the MDPI format, which is used by this journal.

 The disorder prediction of the monomeric high resolution structures is probably an artifact, other methods do not predict them disordered. The other structures are homo trimeric collagen (and a zipper) structures, it is not surprising that they are predicted as disordered on their own.

Reply: Thanks for pointing this out, which has also been noticed by another reviewer. We have mentioned in the Results section that: “It may be possible that the PONDR predictor did not perform well for these sequences, particularly the collagen sequences (Tables S2 and S3)”. However, our suggestion of converting disorder contents to structural entropies would not be affected by the errors induced by predictors, including PONDR.

 The paragraph comparing D and H in the Introduction is welcome, but the question was rather "asking" for a paragraph in Dicussion about general usability of the method.

Reply: Thanks for the suggestion! We added the first paragraph in Discussion (please see the revised manuscript), which reads much better to us.

Round 3

Reviewer 1 Report

The authors answered all of the comments, and this version is acceptable for the publication.